# A Flexible Pressure Sensor Based on Graphene/Epoxy Resin Composite Film and Screen Printing Process

**DOI:** 10.3390/nano13192630

**Published:** 2023-09-24

**Authors:** Qijing Lin, Fuzheng Zhang, Xiangyue Xu, Haolin Yang, Qi Mao, Dan Xian, Kun Yao, Qingzhi Meng

**Affiliations:** 1State Key Laboratory of Mechanical Manufacturing Systems Engineering, Xi’an Jiaotong University, Xi’an 710049, China; qjlin2015@xjtu.edu.cn (Q.L.); xxy04071719@stu.xjtu.edu.cn (X.X.); yanghaolin@stu.xjtu.edu.cn (H.Y.); mq.mq@xjtu.edu.cn (Q.M.); danxian@xjtu.edu.cn (D.X.); vinsent@stu.xjtu.edu.cn (K.Y.); qzmeng2022@xjtu.edu.cn (Q.M.); 2School of Mechanical and Manufacturing Engineering, Xiamen Institute of Technology, Xiamen 361021, China; 3Chongqing Academician Workstation, Chongqing 2011 Collaborative Innovation Centers of Micro/Nano Sensing and Intelligent Ecological Internet of Things, Chongqing Key Laboratory of Micro-Nano Systems and Intelligent Sensing, Chongqing Technology and Business University, Chongqing 400067, China; 4Shandong Laboratory of Yantai Advanced Materials and Green Manufacturing, Yantai 265503, China; 5Xi’an Jiaotong University (Yantai) Research Institute for Intelligent Sensing Technology and System, Xi’an Jiaotong University, Xi’an 710049, China

**Keywords:** flexible pressure sensor, graphene, epoxy resin, piezoresistive principle, abrasive resistance

## Abstract

At present, flexible pressure-sensitive materials generally have problems with weak adhesion and poor wear resistance, which easily result in friction failure when used for plantar pressure detection. In this study, a flexible pressure sensor with the advantages of a wide detection range, fast recovery, and good abrasive resistance was designed and prepared based on the screen printing process. The pressure-sensitive unit with a structural size of 5 mm× 8 mm was a functional material system due to the use of graphene and epoxy resin. The influence of the different mass ratios of the graphene and epoxy resin on the sensing properties was also studied. The test results showed that when the mass ratio of graphene to epoxy resin was 1:4, the response time and recovery time of the sensing unit were 40.8 ms and 3.7 ms, respectively, and the pressure detection range was 2.5–500 kPa. The sensor can detect dynamic pressure at 0.5 Hz, 1 Hz, 2 Hz, 10 Hz, and 20 Hz and can withstand 11,000 cycles of bending. In addition, adhesion tests showed that the high viscosity of the epoxy helped to improve the interlayer bond between the pressure-sensitive materials and the flexible substrate, which makes it more suitable for plantar pressure detection environments, where friction is common.

## 1. Introduction

Gait recognition technology is a new method of disease diagnosis and treatment that can recognize and analyze human gait information [1]. It is helpful to realize the preliminary assessment and judgment of the human health status and has great application potential and value in the fields of disease prediction, rehabilitation medicine, health monitoring, and sports training [2,3,4,5]. The human physiological information that can be used in gait recognition technology is also relatively rich, among which the plantar pressure distribution information, with strong data regularity and easy detection, has a strong correlation with the human gait and is an important information source for gait analysis at this stage [6]. 

Flexible pressure sensors are made of flexible electrodes and functional materials, and they have superior deformability and can be applied to the surfaces of complex structures [7]. According to the difference in the mechanism of the signal being transformed by external excitation, the main types of flexible pressure sensors include capacitive sensors, piezoelectric sensors, triboelectric sensors, and piezoresistive sensors [8]. Among them, capacitive pressure sensors consist of an upper and lower parallel electrode plate, an insulator, and a base. When they are subjected to external pressure, their capacitance will change due to their own deformation, thereby establishing a relationship curve of different pressure and capacitance changes [9]. Liu et al. [10] prepared a flexible capacitive pressure sensor with a crease structure based on carbon nanotubes, whose sensitivity and response time were 2.13 kPa^−1^ and 100 ms, respectively. The sensor had stable sensing performance and can be used for the tactile sensing of soft robots. Hwang et al. [11] used a multistage porous PDMS composite material as the dielectric layer of a capacitive sensor. The prepared sensor had sensitivity of 0.18 kPa^−1^, and it can be used in respiratory detection and other fields. Capacitive sensors have the characteristics of a simple structural design, low power consumption, and low temperature sensitivity, but when they are subjected to external forces, the change in the capacitance is very small (generally at the pF level) [12,13]. Moreover, this type of sensor is highly susceptible to electromagnetic interference, and the signal reading and processing circuits are also relatively complex [14], which makes them unsuitable for plantar pressure detection.

The most common structure of a piezoelectric sensor is a sandwich, wherein the upper and lower structures are the electrode layer and the middle layer is a special dielectric material, such as polyvinylidene fluoride (PVDF) [15], zinc oxide (ZnO) [16], or Pb-based zirconate titanate (PZT) [17]. Its sensing mechanism comes from the piezoelectric effect, i.e., after the specific dielectric is deformed by pressure, an internal polarization effect is generated, and the opposite polarity of the charge is gathered on the two opposite surfaces. Yang et al. [18] modified barium titanate (BaTiO_3_) with polydopamine (PDA) and then mixed it with PVDF to prepare a flexible piezoelectric sensor, which can be used to identify human plantar pressure. The output voltage of the sensor can reach 9.3 V, and the response time is only 61 ms. Wang et al. [19] developed a piezoelectric sensor based on the MXene/PVDF-TrFE dielectric material through the electrospinning process. The sensor has an instantaneous output power of 3.64 mW/m^2^ under the action of 20 N of pressure, which can not only detect pressure but also monitor the changes in humidity. Unlike other sensors, this piezoelectric sensor is passive, so it does not require a power supply, which can reduce the overall power consumption of the sensor system. The sensor also has the advantages of high sensitivity and good dynamic response performance. However, this piezoelectric sensor is very sensitive to noise, and the output signal is very weak [20]. In addition, it has difficulty in detecting static pressure, so its application in the field of plantar pressure detection is limited. As a new wearable electronic and energy-harvesting device, triboelectric sensors have attracted much attention. They have the characteristics of a simple structure, high output voltage, and good biocompatibility [21,22]. Wang et al. [19] designed a triboelectric tactile sensor utilizing a PDMS/eutectic gallium-indium alloy (EGaIn) mixture, which is capable of detecting pressures as low as 0.23 Pa. Qu et al. [23] fabricated a new type of flexible triboelectric tactile sensor by mimicking the fingerprint structure, achieving an impressive response time of only 1.01 ms and enabling functionalities such as password simulation, material recognition, and pulse detection. Nonetheless, triboelectric sensors are very sensitive to ambient humidity, so their packaging requirements are also very strict. This sensor has difficulty in detecting static pressure, and the pressure detection range is narrow; therefore, it is not suitable for the detection of the distribution of plantar pressure [24,25]. 

Compared with the three other types of flexible pressure sensors, flexible piezoresistive sensors have the advantages of stable performance, easy-to-read signals, a low cost, high durability, mass manufacturing, etc. [26,27]. At present, they are among the most widely used flexible sensors in the world and are very suitable for the monitoring of human physiological signals, presenting unparalleled advantages in the field of plantar pressure measurement [28]. Because the conductor material inside flexible piezoresistive sensors is generally a composite, there is a high degree of effort in the selection of the substrate and conductive material. The common base materials include PDMS [29], polyurethane (PU) [30], Ecoflex [31], etc. Conductive materials can generally include carbon nanotubes [32], graphene [33], metal particles [34], etc. In recent years, researchers have also improved the structure of the flexible piezoresistive sensor to improve the sensing performance, such as constructing an array microstructure on the piezoresistive film surface [35] or a bionic mimosa structure [36], and constructing a porous foam at the cost of materials [37]. By coating MXene on the fabric surface, Chen et al. [38] produced a flexible piezoresistive sensor with sensitivity of up to 12.095 kPa^−1^ and a response time of up to 26 ms. Yang et al. [39] designed a pressure-sensitive functional layer with a microsphere top interlocking structure based on polyvinylidene fluoride nanofibers to achieve ultra-high sensitivity of 53 kPa^−1^ within a small pressure range of 58–960 Pa, and it can be attached to human vocal cords and other parts for vibration detection. Although the flexible piezoresistive sensor is suitable for human plantar pressure detection, there are still many problems to be solved. At present, the functional materials of the flexible piezoresistive sensor generally have problems such as weak adhesion and poor abrasive resistance, making them easily worn and causing sensing failure when used for plantar pressure detection. 

In this work, a flexible piezoresistive pressure sensor with the advantages of a wide detection range, fast response/recovery time, and good abrasive resistance was designed and developed based on a graphene/epoxy resin composite film and screen printing process. The effects of different mass ratios of graphene and epoxy resin on the pressure-sensitive properties of the sensor were studied, and the sensor’s sensitivity, response/recovery time, durability, and abrasive resistance were tested. All these test results showed that the sensor has great application potential and value in the field of plantar pressure measurement.

## 2. Materials and Methods

### 2.1. Structure Design and Principal Analysis of the Sensor

The structure of the flexible piezoresistive pressure sensor is shown in Figure 1, and it is composed of a flexible substrate, an interdigital electrode, a composite pressure-sensitive film, and a double-sided tape layer. The sensor’s bottom, intermediate, and top layers are polyimide (PI) substrates with a thickness of 0.13 mm. The PI film (Model PMDA-ODA) used in this study was purchased from Shenzhen Nuoyishun Electronics Co. Ltd. (Guangdong, China), and it has surface energy of 32.4 mJ/m^2^. The low surface energy of the PI substrate allows it to show better compatibility with the pressure-sensitive materials. Furthermore, as the carrier and protective layer of the sensor, PI has the characteristics of a low cost, excellent physical and chemical stability, good flexibility, and so on. The interdigital electrode structure is distributed on the bottom PI substrate and prepared by micro- and nanomanufacturing processes. Cu is used as the interdigital electrode material, with good conductivity and flexibility. The area and thickness of the interdigital electrode are 5 mm × 8 mm and 35 μm, respectively. The distance between the electrodes and the electrode’s width are 0.5 mm. The screen printing process produces the pressure-sensitive film on the interdigital electrode. The different PI substrate layers are bonded by double-sided tape so that the pressure-sensitive film is firmly fixed on the interdigital electrode area.

The flexible pressure-sensitive film is generally a conductive composite material, composed of a flexible matrix material and a conductive material. Commonly, a flexible matrix material is used as the main component, and one or more conductive materials are doped in it so that the electrical properties of the conductive material and the excellent mechanical compliance of the flexible matrix material are combined to show the pressure-sensitive characteristics [40]. In human walking or running, there is friction in addition to the vertical force between the sole and the insole [41]. The frequent action of friction makes it easy for the pressure-sensitive material to fall off the substrate, leading to sensing failure and other problems. In this paper, epoxy resin [42] is selected as the matrix material of the pressure-sensitive film. Its high viscosity can effectively achieve the strong adhesion of the pressure-sensitive material on the substrate, which is conducive to improving the friction resistance and reliability of the sensor. Graphene is a typical two-dimensional carbon-based material, and it has excellent mechanical and electrical properties, a simple preparation process, a low production cost, mass manufacturing, and other advantages [43]. The viscosity of the ink (graphene/epoxy) obtained by the rotary viscometer (Model NDJ-1) is 80,000 mPa·s. Due to the high viscosity of the ink, the ink can be better compatible with the PI substrate. Therefore, graphene is used as a conductive material for the pressure-sensitive film.

As shown in Figure 2, when the graphene/epoxy resin composite film with a regular microstructure is subjected to external pressure, the contact state between the pressure-sensitive composite film and the interdigital electrode will be changed. With the increase in the external force, the contact area between the microstructure peak and the interdigital electrode will gradually increase due to its deformation, which leads to a decrease in the resistance value of the sensor. In addition, the deformation of the composite film will also shorten the spacing between the internal graphene to increase the number of internal conductive paths, which will also lead to a decrease in the resistance value of the sensor. Therefore, the resistance value of the sensor prepared in this paper will decrease when subjected to external pressure, and the morphology size of the surface microstructure and the graphene concentration of the composite film will affect the resistance variation amplitude (i.e., sensitivity) of the sensor.

### 2.2. Preparation and Morphology Characterization of the Sensor

The preparation process of the pressure-sensitive film includes the tape casting method, electrospinning method, transfer printing, and the screen printing process [44,45]. Among them, the screen printing process has the advantages of a low cost, short process cycle, customizable pattern, and mass manufacturing, and iy is widely used in flexible sensor preparation [46]. The flexible graphene/epoxy resin composite film is prepared by the screen printing process. In addition, the conductive paste’s reliable curing also requires a curing agent and diluent. Epoxy resin (model E-51) is an organic polymer that reacts with various curing agents as a base material. Among them, polyether amine (model D400) is a light yellow transparent liquid that shows a good curing effect when combined with epoxy resin. Because the conductive paste is generally viscous, it is not easy to pass through the holes of the screen plate during the screen printing. Therefore, terpinol is used as a diluent to control the viscosity of the conductive paste for the screen printing process.

The screen printing process diagram of the sensor is shown in Figure 3, including the preparation of the conductive paste, the setting of the prefabricated screen plate, scraper scraping, and curing film formation. First, we use an electronic balance to weigh the appropriate amount of terpinol into the beaker, and then add epoxy resin, graphene, and polyether amine to the beaker in turn. A glass rod is used to stir for 15 min so that the graphene is evenly dispersed in the mixture to complete the preparation of the conductive paste. Then, the PI substrate is set on the screen printing table, the prefabricated screen plate is fixed on the PI substrate, and an appropriate amount of conductive paste is applied to the screen plate. The conductive paste on the screen plate is scraped several times by a scraper, which allows the paste to transfer evenly onto the substrate through the mask pattern. During the scraping process, attention should be paid to controlling the angle between the scraper and the screen plate. Regarding the best scraping angle in the screen printing process, Xia P et al. [47] conducted relevant research. When the scraping angle is large, the transfer rate of the conductive ink is slow, resulting in a smaller film thickness for screen printing. When the scraping angle is slight, the corresponding pressure of the scraper is more considerable, and the surface roughness of the printed film is also more significant. The above results will affect the mechanical properties of the pressure-sensitive film. Therefore, the scraping angle should not be too large or too small, so 45° is selected. Finally, the sample is placed in a heating furnace at 110 °C for 60 min to accelerate the solidification of the conductive paste. Then, the preparation of the graphene/epoxy resin composite film is completed. 

According to the conduction path theory and the tunneling current theory [48,49], the mass ratio of graphene in the conductive paste determines the pressure-sensitive characteristics of the sensor. Suppose that the amount of graphene in the conductive paste is too low. In this case, the composite film will resemble an insulating object, which will cause the initial resistance value of the sensor to be too large to be used for pressure detection. On the contrary, if the graphene content is too high, the composite film will be similar to a conductor, which will cause the initial resistance value of the sensor to be too small and it will be unable to respond to external pressure. Therefore, it is necessary to regulate the mass ratio of graphene in the composite film to optimize the pressure-sensitive output characteristics of the sensor. In addition, when preparing the conductive paste, it is also essential to choose a reasonable proportion of epoxy resin and curing agent (i.e., polyether amine). If the amount of polyether amine is insufficient, the reaction between the epoxy resin and the curing agent will be inadequate, and the conductive paste cannot be cured completely. If doped too much, it will cause the conductive paste to leave curing agent residue after the end of the curing reaction, which will also increase the viscosity of the film. However, if the doping amount of polyether amine is too high, it will lead to curing agent residue after the curing reaction, which will also increase the viscosity of the film and cannot be cured. To study the influence of different graphene content on the output characteristics of the sensor, the ratio of graphene and epoxy resin is adjusted to prepare various types of conductive paste, such as 1:3, 1:4, 1:5, 1:6, 1:7, and 1:10. The contents of different components of the conductive paste are shown in Table 1. The mass ratio of epoxy resin to polyether amine follows the principle of 2:1, and the addition of terpinol is approximately 0.3 g, which ensures that the conductive paste is convenient for screen printing and curing molding.

In the screen printing process, the conductive paste is transferred to the PI substrate through the regular arrangement of holes on the screen plate under the action of a scraper. During the screen rebound, the conductive paste will cause the phenomenon of wire drawing due to its viscosity, resulting in the regular micromesh structure on the surface of the cured composite film. The surface morphology of the graphene/epoxy resin composite film is shown in Figure 4a. Different mesh microstructures are arranged regularly, and their sizes are the same, which indicates that the consistency of the composite film prepared by the screen printing process is good. The internal structure of a mesh is shown in Figure 4b. The graphene is evenly dispersed in the composite film, and there is no apparent stacking phenomenon. The surface structure and morphology of four composite films with different proportions of graphene are shown in Figure 4c–f, respectively. It can be seen that the higher the ratio of graphene to epoxy resin, the more pronounced the micromesh structure. The reason is that the main factor affecting the viscosity of the conductive paste is the ratio of graphene to epoxy resin, and the higher the proportion of graphene in the paste, the higher the viscosity, which will lead to the more significant drawing phenomenon to make the micromesh structure more obvious. The surface roughness and thickness of the pressure-sensitive film prepared by the optimal mass ratio (1:4) are characterized by confocal microscopy (Model OLS4000) and field emission scanning electron microscopy (Model SU-8010), respectively. The test results are shown in Figure 5. It can be seen from the figure that the surface roughness and thickness of the pressure-sensitive film are 6.45 μm and 61.85 μm, respectively.

## 3. Results and Discussion

### 3.1. Pressure-Sensitive Performance Test of the Sensor

The pressure-sensitive performance test device of the sensor is shown in Figure 6. The maximum pressure range of the tensile machine (model PT-1198GDO) is 50 N, and its resolution is better than 0.001 N. The Keithley 2450 Source Meter (Produced by Tektronix, Beaverton, OR, USA) is used to connect the positive and negative electrodes of the sensor. The current value through the sensor is measured in a constant voltage manner, and the resistance value of the sensor can be calculated according to Ohm’s law. During the experiment, 0.1 V DC voltage is applied to the sensor through the source meter, and the current value of the conductive path is read. Finally, the output resistance value of the sensor is calculated through the current value. The loading device end of the tensile machine is pasted with a 5 mm × 8 mm elastic pad, which is consistent with the size of the pressure-sensitive film of the sensor. The loading device is slowly lowered at a 1 mm/min speed until the pad is in complete contact with the sensor before applying pressure. During the test, the loading pressure of the sensor is gradually increased from 0 to 20 N, i.e., the maximum loading pressure is 500 kPa.

The test results of the sensor show that the composite film with a mass ratio of 1:3 is similar to the conductor after complete contact with the interdigital electrode. The resistance value of the sensor changes very little under the action of external pressure, which indicates that its sensing performance is located in the conductive region and it is unsuitable for pressure sensing. When the mass ratio of graphene to epoxy resin is 1:10, the composite film is non-conductive, which indicates that the sensor’s sensing performance is in the insulation area, and it is also unable to sense external pressure changes. The output characteristic curves for the other four mass ratios of graphene/epoxy resin composite films are shown in Figure 7a. Among them, when the mass ratio of graphene to epoxy resin is 1:4, the conduction current of the sensor has the most prominent change amplitude and the highest sensitivity, showing the best pressure-sensitive characteristics. Therefore, the sensor based on a mass ratio 1:4 is used in the subsequent performance tests and application experiments. In addition, the sensor can detect pressure in the range of 2.5–500 kPa [50], and it can fully cover the spectrum of human plantar pressure changes.

Sensitivity is an important performance index of pressure-sensitive sensors that reflects the output change caused by the unit input [51]. The sensor’s sensitivity (*S*) calculation method is shown in Equation (1), where ∆*I* is the absolute value of the current change, *I*_0_ is the initial value of the current, and *P* is the external pressure imposed on the sensor.
(1)$$S=\frac{\left|\Delta I/I_{0}\right|}{P}$$

The current signal of the sensor under the action of 2.5 kPa pressure is taken as the initial signal, and the sensitivity curve is drawn, as shown in Figure 7b. The fitting calculation results show that the sensitivity of the sensor is 0.156 kPa^−1^ (*S*_1_), 0.068 kPa^−1^ (*S*_2_), and 0.023 kPa^−1^ (*S*_3_) at 2.5–100 kPa, 100–250 kPa, and 250–500 kPa, respectively, which indicates that the sensor has good output response characteristics. The error bar diagram of the sensor at 100 kPa, 250 kPa, and 500 kPa is shown in Figure 7c. As can be seen from the figure, the error bars of the sensor are shorter, which indicates that the sensor has a smaller output error and better repeatability.

### 3.2. Response/Recovery Time Test of the Sensor

Response time is required for the sensor to obtain a stable output signal under external excitation [52]. For the flexible resistive pressure sensor, the viscoelasticity of the material has a significant influence on the response time [53]. The sensor’s response time in this paper is tested by the shaker test platform, as shown in Figure 8. The test platform consists of a signal generator, a power amplifier, a vibration exciter, a source meter, and a computer. The signal generated by the signal generator is amplified by the power amplifier and transmitted to the vibration exciter. Then, the vibration exciter applies the dynamic load to the sensor to realize the force value and frequency control of the load. 

A square wave signal with an amplitude of 4 N is applied to the sensor by the vibration exciter. In particular, the pressure acting on the sensor is 100 kPa. During the dynamic loading test, a source meter provides a constant voltage to the sensor and records and stores the output signal of the sensor. The response/recovery time curve of the sensor is shown in Figure 9. The response time and recovery time of the sensor are 40.8 ms and 3.7 ms, respectively, which indicates that the sensor has a fast response speed and recovery characteristics. The recovery time of the sensor is much smaller than the response time, and its rapid recovery performance is mainly due to the packaging method. A double-sided tape layer bonds the top layer substrate and the bottom substrate. When the load is removed, the elastic deformation of the composite pressure-sensitive material will disappear quickly with the rapid recovery of the top layer substrate and double-sided tape layer, which causes the contact area between the composite pressure-sensitive material and the interdigital electrode to decrease rapidly. The reduction in the contact area between the two will cause the sensor’s output resistance value to increase rapidly, so the sensor has rapid recovery performance.

### 3.3. Dynamic Response Test of the Sensor

To verify the response of the sensor under different loading frequencies, the sensor is cyclically loaded with different loading frequencies. The loading frequencies are 0.5 Hz, 1 Hz, 2 Hz, 10 Hz, and 20 Hz, respectively. The loading pressure is constant at 125 kPa. The test results are shown in Figure 10, indicating that the output response of the sensor under different loading frequencies is consistent. At the same time, it can be found that with the increase in loading frequency, the output amplitude of the sensor shows a decreasing trend. Since the loading signal of the vibration exciter is a sinusoidal signal, the contact time between the vibration exciter and the sensor becomes shorter with the increase in frequency. When the sensor is not fully responsive, the vibration exciter begins to unload, which leads to a difference in the high- and low-frequency response of the sensor. However, this difference is slight, and the output response is consistent, indicating that it can meet the test requirements of the sensor.

To test the response of the sensor to different pressure amplitudes, dynamic loads of different amplitudes are applied to the sensor by the vibration exciter, and the test results are shown in Figure 11. The output characteristics of the sensor under other loads are good and have good consistency. Moreover, the output response of the sensor under different loading amplitudes is also one-to-one, corresponding to the results of the pressure-sensitive characteristics (see Figure 7a).

### 3.4. Durability and Adhesion Test of the Sensor

Durability determines the life cycle of the sensor and is an essential indicator of sensing performance and economic cost [54]. The durability test curve of the sensor is shown in Figure 12. A sinusoidal load of 125 kPa is applied to the sensor at a frequency of 2 Hz, and the number of cyclic loads exceeds 11,000. The test results show that the sensor still maintains the original output response after 11,000 cycles of loading, and there is no apparent baseline drift phenomenon, which proves that it has good durability.

In the process of walking, there is not only vertical force between the sole and the ground but also a large friction force. When the adhesion between the pressure-sensitive material and the substrate is weak, the pressure-sensitive material can easily peel off, which causes damage to the sensor and affects the detection of the plantar pressure. Based on the (GB/T 9286-1998 (ISO 2409:1992) [55]) paint film grid method [56], the adhesion test of the pressure-sensitive material is carried out in this paper. The surface of the pressure-sensitive material is cut multiple times by a cross-cut tester to form a grid structure, with both cutting directions at an angle of 90°. After using a soft brush to clean up the film area that is scratched, we use adhesive tape to paste this area and slowly peel off the tape. Finally, the film is placed under a microscope to observe the mesh morphology, and the adhesion can be judged by the falling off of the pressure-sensitive material. The surface of the pressure-sensitive material after cutting is shown in Figure 13. The cutting edge of the material is relatively complete and smooth, and there is no case of material falling off. The classification of the adhesion test results of different standards is shown in Table 2. It can be seen from the table that the adhesion level of the printed materials is of the highest level 0 (ISO 2409:1992), i.e., the highest level 5B (ASTM D3359-2017 [57]), indicating that the sensor has abrasive solid resistance. Compared to the method of laser integration of graphene inside materials [58], the excellent adhesion of the pressure-sensitive material in this paper is due to the addition of epoxy resin to the conductive ink.

### 3.5. Performance Comparative Analysis of Different Sensors

The performance comparison of the flexible pressure sensor with other sensors is shown in Table 3. Compared with the pressure sensors of other pressure-sensitive materials, the flexible pressure sensor with the graphene/epoxy composite film has apparent advantages in terms of the pressure measurement range and recovery time. Although the pressure sensor with a conductive elastomer film developed by Zhong et al. has a maximum pressure detection range of 600 kPa, the recovery time of the sensor is much longer than that of the graphene/epoxy sensor. In addition, few flexible pressure sensor studies focus on their adhesion performance. The flexible pressure sensor proposed in this paper has good adhesion, and its adhesion grade is the highest (ASTM-5B).

## 4. Conclusions

In summary, based on the screen printing process, a flexible graphene/epoxy resin-based pressure sensor for the resistance strain effect is proposed and designed in this paper, which has the advantages of a wide detection range, fast response/recovery, good durability, and wear resistance. The effect of the ratio of graphene in the pressure-sensitive material on the sensing performance is studied and analyzed. The comparative experimental results show that the pressure-sensitive performance of the sensor is the best when the mass ratio of graphene to epoxy resin is 1:4. The sensitivity, response/recovery time, dynamic response, durability, and wear resistance of the sensor with this mass ratio are tested. The test results show that the sensor has a wide range of 2.5–500 kPa, a response time of 40.8 ms, and a recovery time of 3.7 ms, and it can timely respond to the dynamic pressure at 0.5 Hz, 1 Hz, 2 Hz, 10 Hz, and 20 Hz frequencies. The sensor can also withstand 11,000 cycles of loading, indicating that it has good durability. Moreover, the functional layer formed by graphene and epoxy resin has a solid interlayer bonding force with the substrate, and the sensor has a certain anti-wear and anti-failure ability, indicating that it is suitable for the detection of plantar pressure.

## Figures and Tables

**Figure 1 nanomaterials-13-02630-f001:**
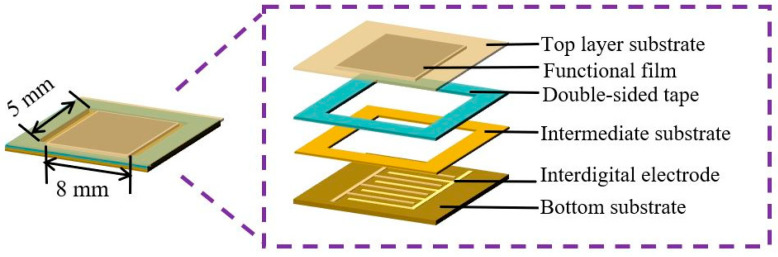
Structure diagram of the sensor.

**Figure 2 nanomaterials-13-02630-f002:**
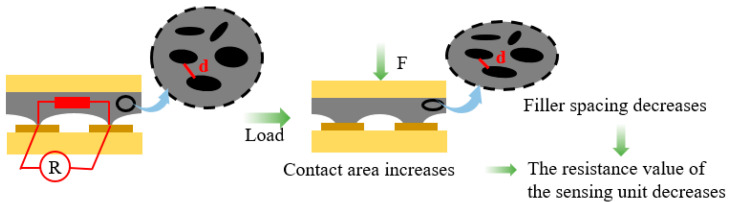
Schematic diagram of the sensor’s pressure-sensitive principle.

**Figure 3 nanomaterials-13-02630-f003:**
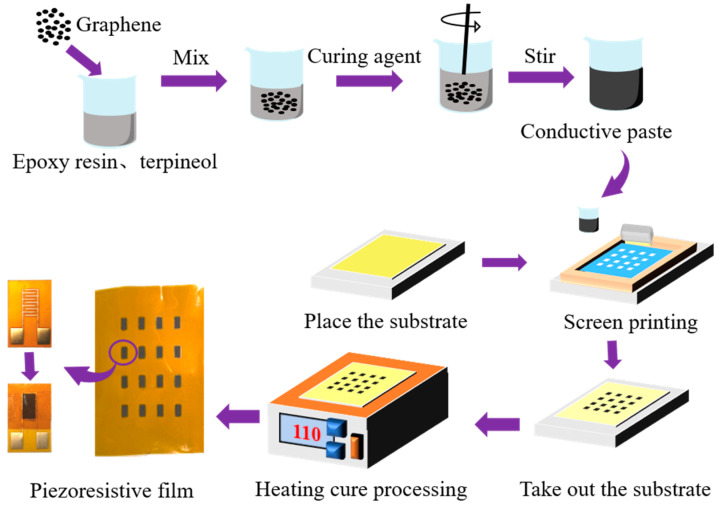
Screen printing process diagram of the sensor.

**Figure 4 nanomaterials-13-02630-f004:**
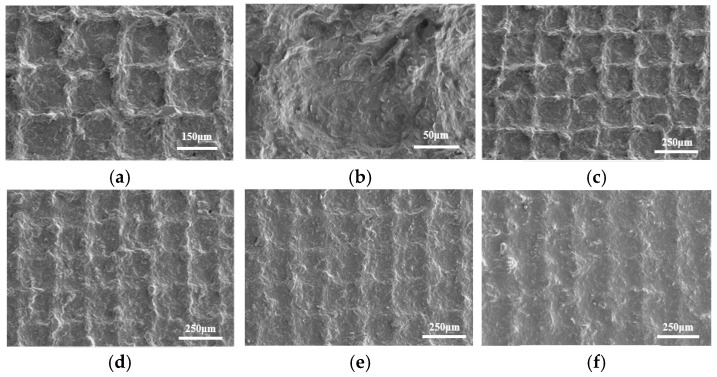
Surface structure and morphology of composite films with different mass ratios of graphene and epoxy resin: (**a**) mesh microstructure (1:4); (**b**) single mesh microstructure (1:4); (**c**) 1:4; (**d**) 1:5; (**e**) 1:6; (**f**) 1:7.

**Figure 5 nanomaterials-13-02630-f005:**
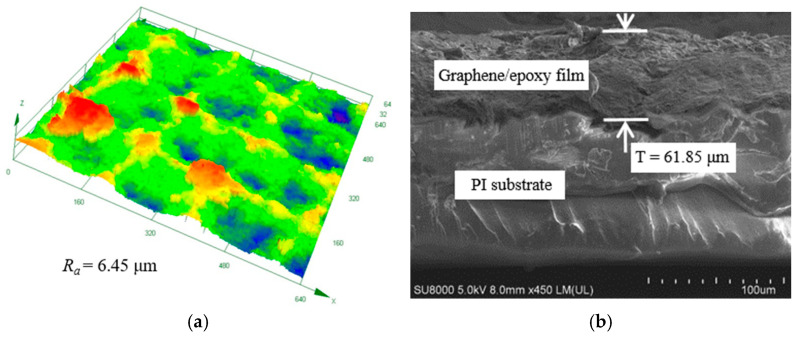
Surface roughness and film thickness of the pressure-sensitive film (mass ratio is 1:4): (**a**) surface roughness; (**b**) film thickness.

**Figure 6 nanomaterials-13-02630-f006:**
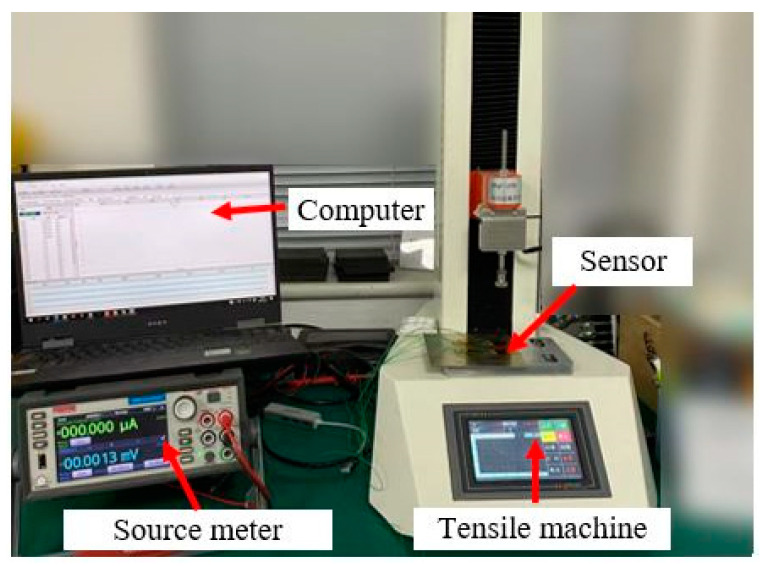
Pressure-sensitive performance test device of the sensor.

**Figure 7 nanomaterials-13-02630-f007:**
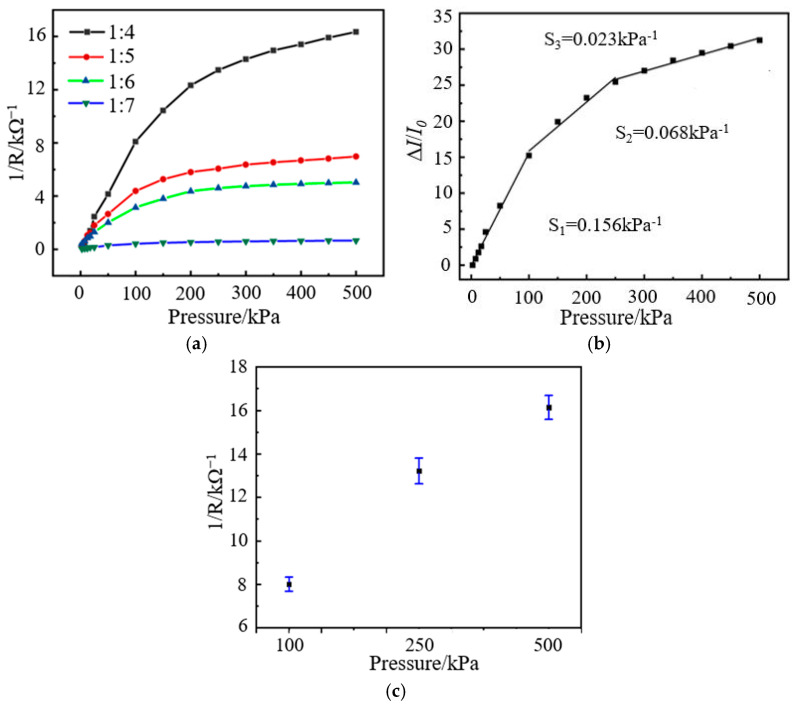
Pressure-sensitive performance test results of the sensor: (**a**) pressure-sensitive characteristic curves of different composite films; (**b**) sensitivity test curve of the sensor (mass ratio is 1:4); (**c**) error bars of the sensor at 100 kPa, 250 kPa, and 500 kPa.

**Figure 8 nanomaterials-13-02630-f008:**
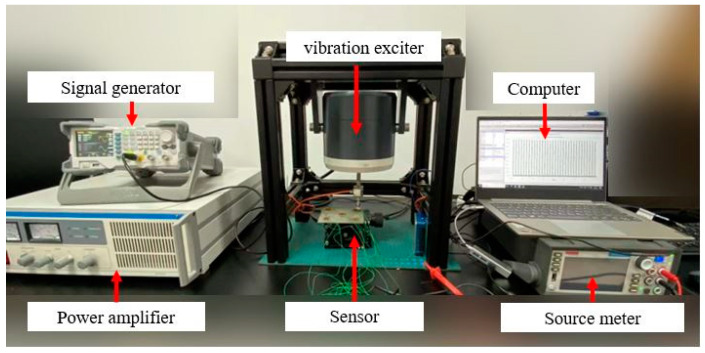
Circuit diagram of row and column multiplexing.

**Figure 9 nanomaterials-13-02630-f009:**
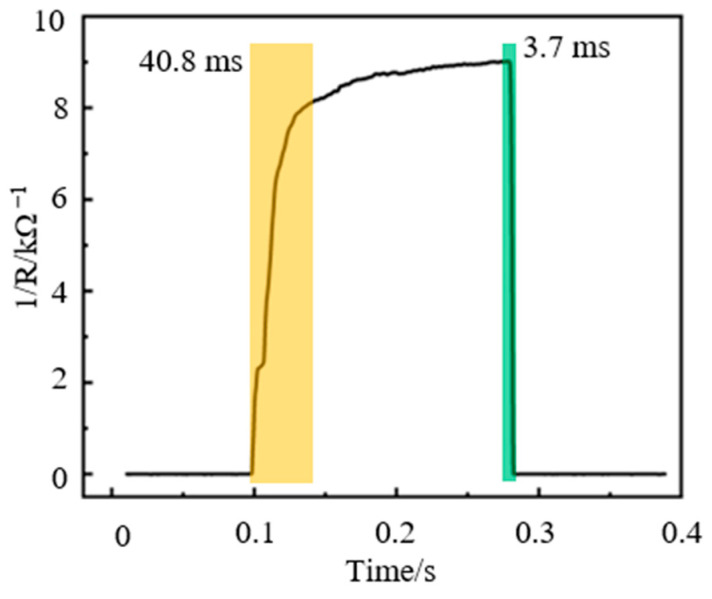
Response/recovery time curve of the sensor.

**Figure 10 nanomaterials-13-02630-f010:**
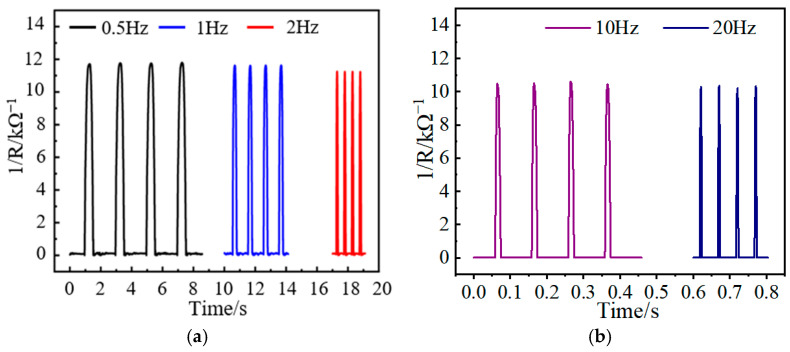
Dynamic response test of the sensor under different loading frequencies: (**a**) 0.5 Hz, 1 Hz, and 2 Hz; (**b**) 10 Hz and 20 Hz.

**Figure 11 nanomaterials-13-02630-f011:**
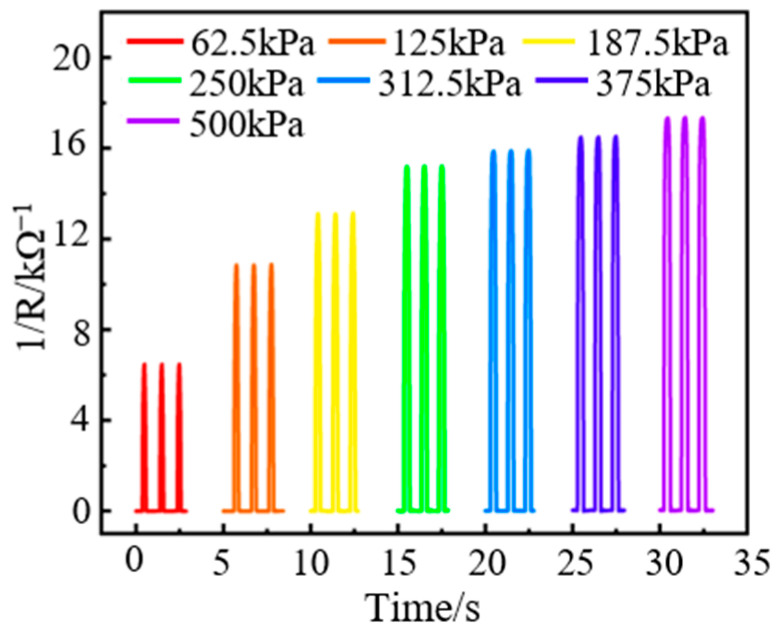
Dynamic response test of the sensor under different loading amplitudes.

**Figure 12 nanomaterials-13-02630-f012:**
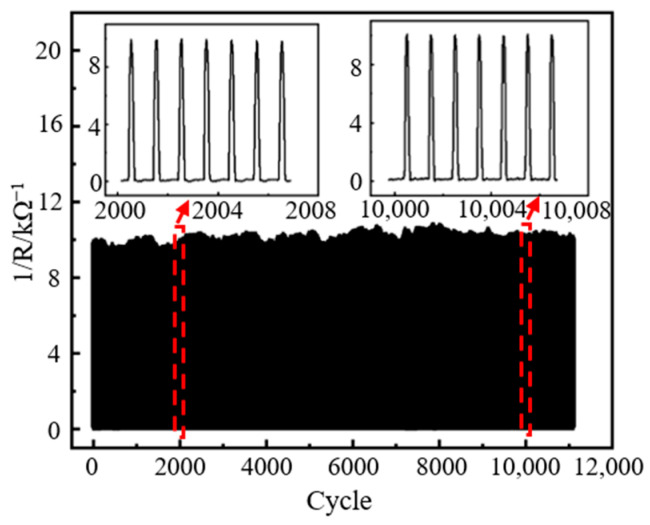
Durability test curve of the sensor.

**Figure 13 nanomaterials-13-02630-f013:**
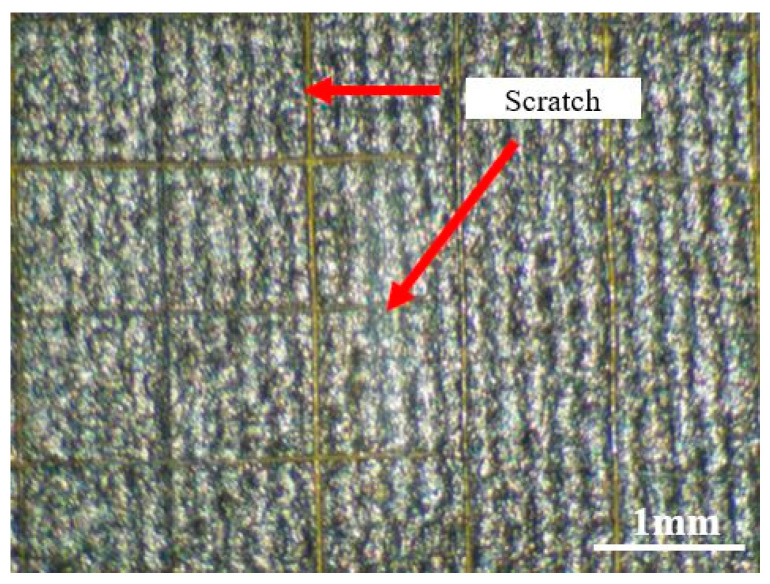
Adhesion test curve of the sensor.

**Table 1 nanomaterials-13-02630-t001:** Comparison of amount of each component in conductive paste.

Mass Ratio	Graphene/g	Epoxy Resin/g	Polyether Amine/g	Terpinol/g
1:3	0.108	0.294	0.165	0.316
1:4	0.108	0.415	0.214	0.319
1:5	0.099	0.515	0.240	0.333
1:6	0.103	0.592	0.310	0.344
1:7	0.110	0.710	0.379	0.295
1:10	0.110	1.018	0.520	0.314

**Table 2 nanomaterials-13-02630-t002:** Classification of adhesion test results of different standards.

Classification	GB/T 9286-1998 (ISO 2409:1992)	ASTM D3359-2017	Description
1	0	5B	Edges of the cuts are completely smooth; none of the squares of the lattice is detached
2	1	4B	A cross-cut area not greater than 5% is affected
3	2	3B	A cross-cut area greater than 5%, but not greater than 15%, is affected
4	3	2B	A cross-cut area greater than 15%, but not greater than 35%, is affected
5	4	1B	A cross-cut area greater than 35%, but not greater than 65%, is affected
6	5	0B	Any degree of flaking that cannot even be classified by classification 4

**Table 3 nanomaterials-13-02630-t003:** Performance comparison table of the flexible pressure sensor with other sensors.

Material	Pressure Range/kPa	Response Time/ms	Recovery Time/ms	Adhesion	Ref.
3D PVDF-Hep/PEDOT nanofibers	0–5	400	200	/	[59]
P-HCF	0.02–600	40	20	/	[6]
rGO-PDMS	>200	15	20	/	[60]
MXene/cotton fabric	0–160	50	20	/	[61]
MXene-textile	29–40	26	50	/	[38]
Graphene/epoxy	2.5–500	40.8	3.7	ASTM-5B	This work

## Data Availability

Data sharing is not applicable to this article.

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
