# Peer review of "A Flexible Pressure Sensor Based on Graphene/Epoxy Resin Composite Film and Screen Printing Process"

_nanomaterials, 2023, doi:10.3390/nano13192630_

Round 1

Reviewer 1 Report

This manuscript reports on the development of a flexible pressure sensor based on graphene/epoxy resin composite film and screen-printing process. My conclusion is that this manuscript cannot be accepted for publication in its current state due to the following reasons:

1) The pre-print characterization of the functional ink and the substrate is missing. For example, what is the surface energy of the substrate (PI)? What is the surface tension and viscosity of the ink (graphene/epoxy) that was printed? How did the authors conclude that the ink is compatible with the PI substrate without these characterizations? These parameters affect the proper printing and adhesion of the electrodes to the substrates. The authors should include this information for better understanding of the work.

2) Post-print characterization should be performed. This includes measurement of thickness and roughness of the printed layer.

3) The authors should include results of an ASTM based adhesion test to demonstrate any peel-off of the printed materials.

4) The authors state that the best scraping angle was 45 degrees. The authors should characterize and provide results to show the reasoning for this conclusion. For example, was it based on the roughness of the printed layer or was it based on uniform thickness of the printed layer?

5) The authors state that the pressure range of 2.5 kPa – 500 kPa fully covers the range of human plantar pressure change. The authors should include a reference for this statement.

6) How many pressure sensors were tested for each of the tests/characterizations to demonstrate repeatability? Error bars should be included for each result.

7) The dimensions of the interdigital electrodes should be included.

8) The photographs of the fabricated interdigital electrodes and square electrode should be included.

9) The authors should include a table comparing the performance of this work to the state-of-the-art in printed pressure sensors.

Author Response

Thank you very much for your professional suggestion. Please see the attached reply document.

Reviewer 2 Report

The work is relevant and is of undoubted interest to readers. I would like to see answers to the following questions in the article:

1. To what extent is the adhesion of graphene by screen printing process better than obtained by the method of laser integration of graphene inside materials (recent works by Raul D. Rodriguez, https://orcid.org/0000-0003-4016-1469)?

2. Why are frequencies above 2 Hz not considered? Sensors with frequencies of 10-20 Hz could significantly expand the scope of the technology.

Author Response

(The authors gave the same response as above.)

Round 2

Reviewer 1 Report

The authors have satisfactorily addressed all the comments.

English editting is required.